# Common Extensor Complex Is a Predictor to Determine the Stability in Simple Posterolateral Elbow Dislocation: Analysis of MR Images of Stable vs. Unstable Dislocation

**DOI:** 10.3390/jcm9103094

**Published:** 2020-09-25

**Authors:** Chul-Hyun Cho, Beom-Soo Kim, Jaehyuck Yi, Hoseok Lee, Du-Han Kim

**Affiliations:** 1Department of Orthopedic Surgery, School of Medicine, Keimyung University, Daegu 42601, Korea; oscho5362@dsmc.or.kr (C.-H.C.); BSKim@dsmc.or.kr (B.-S.K.); 2Department of Radiology, School of Medicine, Keimyung University, Daegu 42601, Korea; yijh72@gmail.com; 3Department of Radiology, School of Medicine, Kyungpook National University, Daegu 41566, Korea; cathrad@hananet.net

**Keywords:** elbow, dislocation, ligament, tendon, tear, MRI

## Abstract

Simple posterolateral elbow dislocations (SPLED) may be treated nonoperatively using closed reduction, followed by controlled mobilization. However, the extent of soft tissue injuries might affect the choice of treatment, rehabilitation approach, and prognosis. The purpose of this study is to compare the characteristics of soft tissue injuries between patients with unstable and stable SPLED using MRI findings. Thirty MRIs of elbows with SPLED (unstable group (*n* = 15); stable group (*n* = 15)) were randomly reviewed by two experienced musculoskeletal radiologists. Soft tissue injuries were characterized as an intact, partial tear or complete tear for the: medial collateral ligament (MCL) complex, common flexor complex, lateral collateral ligament (LCL) complex, common extensor complex, anterior capsule, and posterior capsule. Moderate to substantial interobserver reliability and substantial to perfect intraobserver reliability were observed for medial and lateral complexes in SPLED. The proportion of soft-tissue injuries of the common extensor complex were significantly different between the unstable (four partial tears and 11 complete tears) and stable groups (11 partial tears and four complete tears). In conclusion, based on MRI findings, the degree of common extensor complex injuries may be a predictor of stability and help inform treatment decisions for SPLED.

## 1. Introduction

The elbow joint is the second most common site of dislocation in humans, and more than 90% of simple elbow dislocations (i.e., no relevant osseous lesions) occur in the posterolateral and posterior direction [1,2,3,4]. Simple posterolateral elbow dislocations (SPLED) are usually stable after reduction and have been treated nonoperatively, followed by controlled mobilization, with favorable results [5,6]. However, several investigators have emphasized that these injuries are not entirely benign [1,7,8].

Elbow dislocations are particularly damaging for soft tissues (e.g., ligaments, capsules, tendons). Various imaging modalities (e.g., plain radiographs, stress radiographs, ultrasound, computed tomography (CT), magnetic resonance imaging (MRI)) can be used to evaluate the degree of soft tissue injury and elbow instability [9,10,11]. However, dependent on the patient’s compliance, these tests may not elicit instability [12]. MRI, a commonly used non-invasive test for assessing lesions of the soft tissues of the elbow, is highly accurate and remains the gold standard imaging approach to characterize the extent of soft tissue injuries. However, a standardized diagnostic algorithm for acute simple elbow dislocations has not yet been established. Additionally, there has been little documentation of MRI analysis of predictors to determine the stability in SPLED.

The aim of this study was to compare the characteristics of soft tissue injuries in patients with unstable SPLED and those with stable dislocations using MRI findings. This study was conducted to confirm the hypothesis that the status of the common extensor complex is a predictor to determine stability in SPLED.

## 2. Material and Methods

This retrospective cohort study included 30 patients with SPLED who were managed at our tertiary care hospital between January 2009 and September 2018 (Figure 1).

The included patients were divided into two groups based on the stability of their injury (unstable group (*n* = 15); stable group (*n* = 15)) and matched for age and sex. Inclusion criteria included: (1) posterolateral elbow dislocation as confirmed with the use of radiographs at the time of the initial injury, (2) referral ≤1 week after the initial injury, (3) no fracture (except avulsion fracture of coronoid process tip). Exclusion criteria were: (1) chronic dislocation, (2) any other history of elbow surgery or trauma, and (3) osteoarthritic elbow. Institutional review board approval was obtained for this study (IRB No. 2020-01-061), and informed consent was obtained from all individual participants included in the study.

All patients were initially managed with closed reduction under sedation in the emergency department. Whenever possible, an attempt was made to evaluate elbow stability during the full range of motion after closed reduction. If the elbows were stable following closed reduction, plaster immobilization was completed for 10 days to three weeks, followed by a gentle passive range of motion exercises (stable group). Indications for surgical procedure included: (1) failed closed reduction, (2) elbow subluxation or a non-congruent joint following closed reduction confirmed on radiographs, (Figure 2), and (3) re-dislocation or gross laxity on physical examination following closed reduction (unstable group).

### 2.1. Surgical Treatment

In examination under anesthesia, a valgus stress test was performed initially at 30°–40° of elbow flexion with the forearm in pronation. In patients with a stable medial side during the valgus stress test, usually only the medial collateral ligament (MCL) was torn without rupture of the overlying common flexor complex, and only the lateral collateral ligament (LCL) complex was repaired. The lateral complex repair was performed using a modified Kocher posterolateral approach. The torn edge of the LCL complex and common extensor complex was tagged and reattached to the isometric point of the lateral epicondyle using a suture anchor (Figure 3). In the absence of a firm endpoint or dislocation during the valgus stress test, the MCL was repaired first. MCL complex and flexor-pronator groups were reattached to the isometric point of the medial epicondyle using a suture anchor. The varus and valgus stability of the elbow joint was then re-evaluated, and no LCL complex repair was performed if a stable elbow joint was obtained. If dislocation recurred after MCL repair, additional repair of the LCL complex was performed.

### 2.2. Radiological and MRI Analysis

Simple radiographs were initially performed before and after reduction. MRI scans were undertaken to evaluate injury patterns of soft tissue and bone, including intra-articular damage. All MRIs were performed using 1.5 Tesla scanner (Siemens Magnetom Avanto System; Siemens Medical, Erlangen, Germany) with dedicated elbow specific surface coils. We obtained MR T1/T2-weighted coronal, sagittal, and axial images and T2-weighted fat-suppression images in at least one plane. Special MRI reconstructions (e.g., coronal oblique images) were not conducted. Evaluation of MRI scans was conducted on a medical viewing monitor with adjustable brightness and contrast control [13,14]. The average time from initial injury to MRI imaging was 1.5 days (range 0–4 days). Two experienced musculoskeletal radiologists assessed blinded images for the intensity and morphology of elbow soft tissues. Soft tissue injuries were characterized as intact, partial tear, or complete tear for the: (1) MCL complex, (2) common flexor complex, (3) LCL complex, (4) common extensor complex, (5) anterior capsule and (6) posterior capsule. Partial-thickness tears were characterized by abnormal ligamentous morphology and signal intensity on fluid-sensitive sequences, with full-thickness tears demonstrating an area of complete discontinuity along the course (Figure 4). Three months later, the radiologists repeated a randomized analysis to evaluate interobserver reliability.

### 2.3. Statistical Analysis

The independent *t*-test and the chi-square test were used to compare baseline demographics data between the two groups. Intra-observer reliability and inter-observer reliability were assessed by calculating the κ correlation coefficient (with 1.0 representing total agreement and 0 representing no agreement) [15]; κ coefficients interpretation was performed using the Landis and Koch criteria [16]. They defined a κ value of >0.8 as “almost perfect agreement”, between 0.6 and 0.8 as “substantial agreement”, between 0.4 and 0.6 as “moderate agreement”, between 0.2 and 0.4 as ”fair agreement”, and <0.2 as “slight agreement”. The paired Student *t*-test was used to identify potential statistical differences between the mean κ values. *p* values of <0.05 were considered significant.

## 3. Results

The mean age of patients was 53.7 years (range 37–73 years). Nineteen patients (63.3%) were men, and 11 (36.7%) were women. Elbow dislocation occurred on the right side (*n* = 17; 56.7%). Patient demographics are summarized in Table 1, and there were no significant differences between the groups in terms of age, sex, involved side, and time between trauma and imaging (all *p* > 0.05).

### 3.1. Interobserver and Intraobserver Reliability

As presented in Table 2 and Table 3, the interobserver and intraobserver reliability of the structures was as follows: (1) MCL complex (κ = 0.591 and 0.862), (2) common flexor complex (κ = 0.466 and 0.683), (3) LCL complex (κ = 0.782 and 0.782), (4) common extensor complex (κ = 0.435 and 0.861), (5) anterior capsule (κ = 0.094 and 0.887), (6) posterior capsule (κ = 0.122 and 0.774).

### 3.2. Unstable Group vs. Stable Group

In the unstable group, a partial tear of the MCL complex was found in four cases (26.7%), and the total tear was 11 (73.3%). Additionally, there was one (6.7%) intact without common flexor complex tear, six (40%) partial tears, and eight (53.3%) complete tears. On the lateral side, a complete tear of the LCL complex was observed in all 15 cases, and the common extensor complex was completely torn in 11 cases (73.3%) and partially torn in four cases (26.7%). The anterior capsule was completely torn in five cases (33.3%) and partially torn in 10 (66.7%) cases. The posterior capsule was completely torn in 12 cases (80%) and partially torn in three cases (20%).

In the stable group, a partial tear of the MCL complex was found in four cases (26.7%), and the total tear was 11 (73.3%). Additionally, there were five (33.3%) intact cases without common flexor complex tear, six (40%) partial tears, and four (26.7%) complete tears. On the lateral side, a complete tear of LCL complex was observed in 13 cases (86.7%) and a partial tear in two cases (13.3%) and common extensor complex was completely torn in four cases (26.7%) and partially torn in 11 cases (73.3%). The anterior capsule was completely torn in three cases (20.0%) and partially torn in 12 (80.0%) cases. The posterior capsule was completely torn in six cases (40%) and partially torn in nine cases (60%).

The proportion of common extensor complex injuries were significantly different between the unstable group (four partial tears and 11 complete tears) and stable group (11 partial tears and four complete tears) (*p* = 0.028). There were no significant differences between the groups in terms of the proportion of injuries for the MCL complex, common flexor complex, LCL complex, anterior capsule, and posterior capsule (*p* > 0.05) (Table 4).

## 4. Discussion

The present study revealed moderate to substantial interobserver reliability and substantial to perfect intraobserver reliability for medial and lateral soft tissues in SPLED. It is particularly interesting to note that the common extensor complex was more commonly injured in the unstable group compared with the stable group.

Although MRI has been shown to be excellent at detecting ligament injuries in a cadaveric model of chronic elbow instability [17,18,19], there are few studies with inconsistent results for acute injuries. In acute elbow dislocations, most patients are not able to fulfill full extension of the elbow, thus making interpretation much more difficult because the collateral ligaments are not tensioned, and MRI reconstructions are not uniform. In addition, when elbow dislocation occurs, the joint capsule is also damaged in many cases. In our study, the anterior and posterior capsules were completely or partially torn in all 30 cases. An occurrence that leads to a dissection of joint fluid through the soft tissue planes of the forearm and that abnormal fluid in the soft tissues may affect MRI analysis, such as common extensor complex or common flexor complex [20]. For these reasons, interpreting MRI scans after acute elbow dislocation might be difficult. Schnetzke et al. [14] conducted a study that assessed the interobserver and intraobserver agreement on ligamentous injuries on MRI in acute simple elbow dislocation. In their study, interobserver agreement was fair to moderate for collateral ligaments (LCL: 0.441, MCL: 0.275), and the assessment of extensor and flexor tendon injuries showed slight interobserver agreement (extensor: 0.049, flexor:0.143).

Numerous studies have reported favorable functional results following simple elbow dislocation, regardless of the treatment approach used [1,2,21]. However, approximately 30% of elbows with simple dislocations were easily re-dislocated in the semi-flexion or extension position and more than 50% of patients with simple elbow dislocations felt residual pain or stiffness [7,22]. Several authors proposed that there are some instances in which simple elbow dislocations are not effectively treated nonoperatively [22,23]. Therefore, a systematic surgical approach should be considered. However, it is not yet clear which cases should be treated with which operative approach and few studies have analyzed the predictive factors of elbow instabilities as a consequence of dislocations. The incongruity of the elbow joint in MRI or CT can indicate an indirect sign of elbow instability. In 2015, Hackl et al. [2] provided MRI criteria indicative of posterolateral rotatory instability signs. Additionally, they suggested that cutoff points of 1.2 mm for radio-capitellar joint incongruence in the sagittal plane and a 0.7 mm ulnar–humeral incongruence in the axial plane are suitable to screen for posterolateral rotatory instability.

Accurately characterizing injuries to musculotendinous structures or capsules is important after elbow dislocation because these structures are significant active stabilizers of the elbow [2,24]. The extent of individual injuries is thought to be dependent on the energy expended and the degree of displacement. Luokkala et al. [25] evaluated 17 consecutive cases of stable simple elbow dislocations and evaluated similar soft tissue structures as this study: MCL, flexor-pronator muscle mass origin, anterior capsule, posterior capsule, LCL, and extensor muscle mass origin. In contrast to this study, however, complete anterior capsule tears were most common (12/17), followed by MCM and LCL tears (10/17, 9/17). Only two patients had complete ruptures of the common flexor or posterior capsule, and the only patient with a posterior capsule tear in their study was also the only patient with complete disruption of the extensor muscle mass. The authors proposed that this patient had the highest energy injury and the greatest risk of recurrent instability. In a subgroup analysis conducted here, a complete tear of the posterior capsule and common extensor complex was observed in 12 (80%) and 11 patients (73.3%), respectively. Although these were no significant differences observed between the groups with respect to the proportion of injuries to the posterior capsule, an accurate assessment of this structure is important to determine the degree of trauma energy after SPLED. As mentioned earlier, this study revealed that the anterior and posterior capsules were completely or partially torn in all 30 cases. Unlike a ligament-like structure, joint capsules have a wide and variable origin and insertional attachments. Additionally, because it can be variously observed based on the position of the elbow joint, it is difficult to diagnose a rupture using MRI and to obtain a high degree of interobserver and intraobserver agreement.

There were several limitations to this study. First, this is a retrospective study, meaning that only a small proportion of patients in the series had an MRI examination and that there was a potential risk of selection bias. Secondly, MRI only provides a static image whereas elbow instability is dynamic; correlations with a dynamic test (e.g., stress test) were not made. Thirdly, MRI without coronal oblique reconstructions was used in all patients, had this been used, the quality of MRI may have improved. Lastly, the assessment was done by only two radiologist examiners, and thus the findings may not necessarily be generalizable.

In conclusion, moderate to substantial interobserver reliability and substantial to perfect intraobserver reliability were observed for medial and lateral complexes in SPLED. Based on MRI findings, the degree of common extensor complex injury may be a predictor of stability and help inform treatment decisions for SPLED. Therefore, we recommend that surgical procedures, or delayed rehabilitation, should be considered in patients with these injuries.

## Figures and Tables

**Figure 1 jcm-09-03094-f001:**
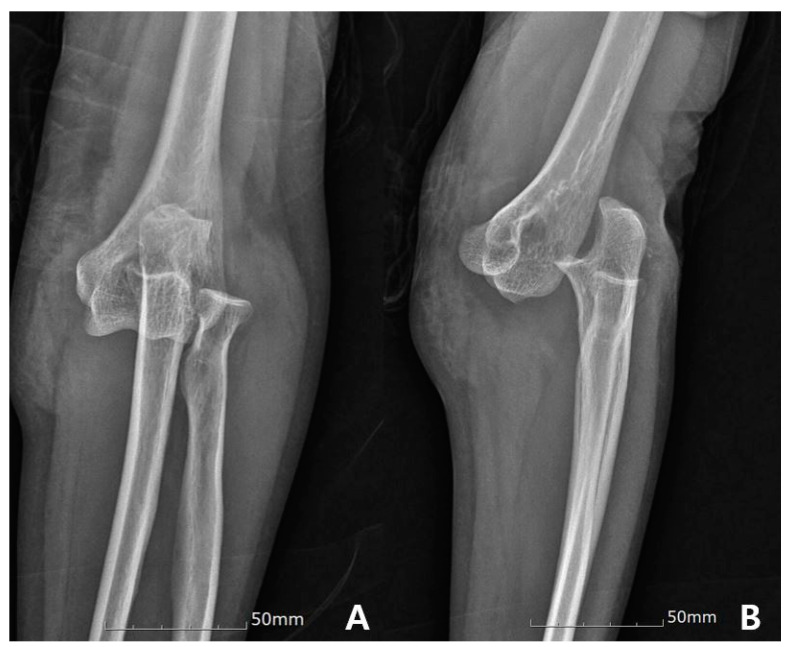
A 53-year old woman injured by slip down. Anteroposterior (**A**) and lateral (**B**) radiographs show simple posterolateral elbow dislocation.

**Figure 2 jcm-09-03094-f002:**
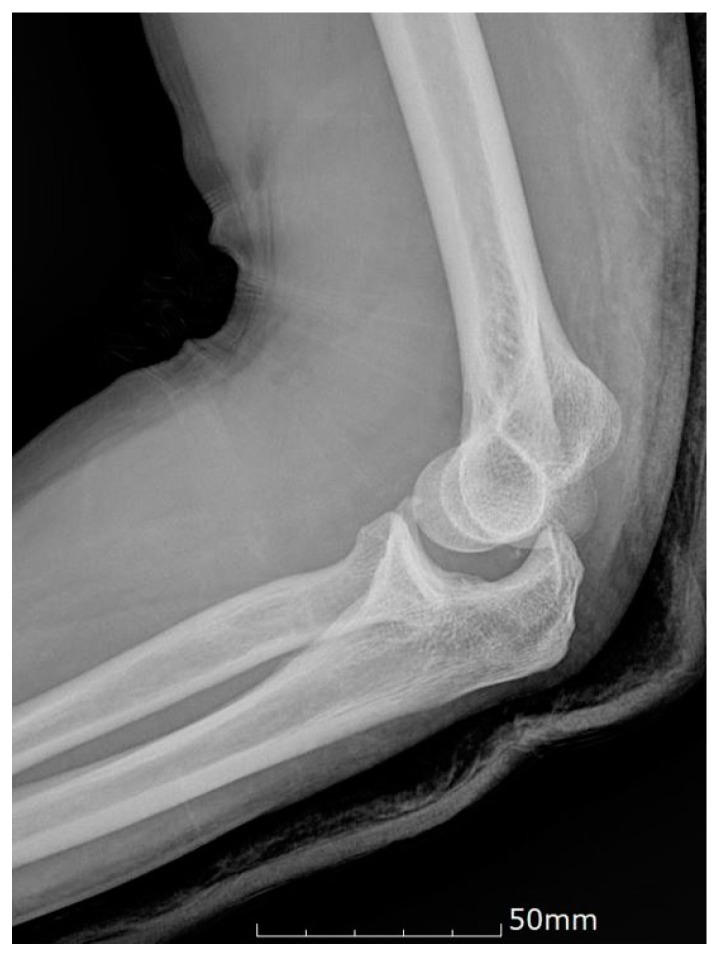
Post-reduction lateral radiograph reveals incongruence at the ulno-humeral articulation.

**Figure 3 jcm-09-03094-f003:**
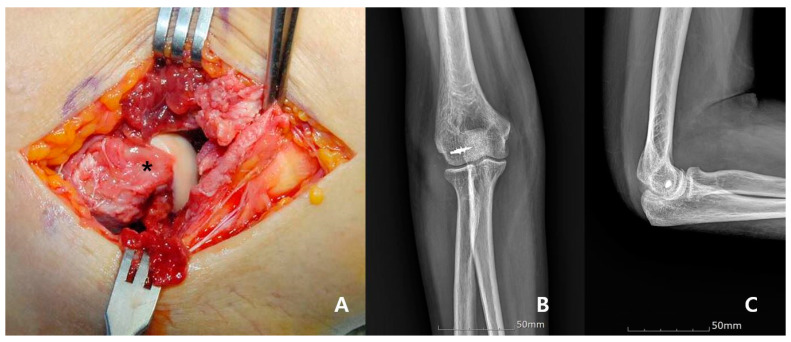
A 59-year old woman treated by repair of LCL alone. Intraoperative photograph reveals that the LCL and common extensor complex had a distractive tear pattern and were retracted from the lateral epicondyle (asterisk) (**A**). The lateral collateral ligament and common extensor complex are repaired with suture anchors; postoperative anteroposterior (**B**), and lateral (**C**) radiograph.

**Figure 4 jcm-09-03094-f004:**
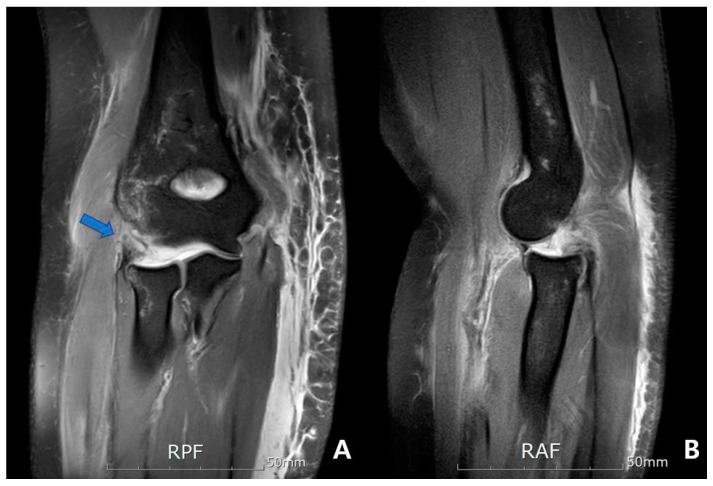
A 59-year old woman injured by a fall from height. Magnetic resonance images reveal stripping-type complete tears of the lateral collateral ligament and common extensor complex (arrow) (**A**) and a slightly posterior subluxated radial head (**B**).

**Table 1 jcm-09-03094-t001:** Demographic data.

Parameter	Unstable Group	Stable Group	*p* Value
Age (year) (SD)	54.0 (10.3)	53.3 (8.3)	0.420
Sex (*n*)			0.864
Male	9	10	
Female	6	5	
Involved side (*n*)			0.749
Right	9	8	
Left	6	7	
Time between trauma and imaging (day) (SD)	1.4 (2.2)	1.5 (3.0)	0.689

SD: standard deviation.

**Table 2 jcm-09-03094-t002:** Interobserver agreement between radiologist 1 and 2 per injured structures.

Injured Structures	First-Round	Second-Round	Mean κ-Value
MCL complex	0.627	0.556	0.591
Common flexor complex	0.550	0.383	0.466
LCL complex	0.782	0.782	0.782
Common extensor complex	0.487	0.383	0.435
Anterior capsule	0.125	0.063	0.094
Posterior capsule	0.148	0.097	0.122

Κ: Kappa; MCL: medial collateral ligament; LCL: lateral collateral ligament.

**Table 3 jcm-09-03094-t003:** Intraobserver agreement between radiologist 1 and 2 per injured structures.

Injured Structures	Radiologist 1	Radiologist 2	Mean κ-Value
MCL complex	0.911	0.814	0.862
Common flexor complex	0.567	0.798	0.683
LCL complex	0.782	0.782	0.782
Common extensor complex	0.798	0.923	0.861
Anterior capsule	0.918	0.857	0.887
Posterior capsule	0.789	0.760	0.774

Κ: Kappa; MCL: medial collateral ligament; LCL: lateral collateral ligament.

**Table 4 jcm-09-03094-t004:** Incidence of injured structures in unstable and stable groups.

Injured Structures	Unstable Group (*n* = 15)	Stable Group (*n* = 15)	*p* Value
MCL complex			1.000
Intact	0 (0%)	0 (0%)	
Partial tear	4 (26.7%)	4 (26.7%)	
Complete tear	11 (73.3%)	11 (73.3%)	
Common flexor complex			0.135
Intact	1 (6.7%)	5 (33.3%)	
Partial tear	6 (40.0%)	6 (40.0%)	
Complete tear	8 (53.3%)	4 (26.7%)	
LCL complex			0.464
Intact	0 (0%)	0 (0%)	
Partial tear	0 (0.0%)	2 (13.3%)	
Complete tear	15 (100.0%)	13 (86.7%)	
Common extensor complex			0.028 *
Intact	0 (0%)	0 (0%)	
Partial tear	4 (26.7%)	11 (73.3%)	
Complete tear	11 (73.3%)	4 (26.7%)	
Anterior capsule			0.680
Intact	0 (0%)	0 (0%)	
Partial tear	10 (66.7%)	12 (80.0%)	
Complete tear	5 (33.3%)	3 (20.0%)	
Posterior capsule			0.062
Intact	0 (0%)	0 (0%)	
Partial tear	3 (20.0%)	9 (60.0%)	
Complete tear	12 (80.0%)	6 (40.0%)	

*: statistically significant. MCL: medial collateral ligament; LCL: lateral collateral ligament.

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
