# Peer review of "Common Extensor Complex Is a Predictor to Determine the Stability in Simple Posterolateral Elbow Dislocation: Analysis of MR Images of Stable vs. Unstable Dislocation"

_jcm, 2020, doi:10.3390/jcm9103094_

Round 1
Reviewer 1 Report
Dear Authors
I realize that authors have many journals to consider when they want to publish their work, so I appreciate your interest in JCM;
It is evident that you have put a great deal of effort into this project and I want to praise your efforts.
The manuscript as currently written suggests that it might be suitable for sharing information about this particular clinical condition.
The manuscript need the minor revisions:
- material and methods section: insert the number of Institutional review board approval
- statistical analysis section: if p values of < 0.05 were considered significant, highlight them in some way in the tables (e.g. in bold).
I should like to thank you for giving me an opportunity to consider this work for publication.
Author Response
Response to Reviewer 1 Comments
Point 1: The manuscript need the minor revisions:
- material and methods section: insert the number of Institutional review board approval
Response 1: Thanks for your comment. We added a number of IRB. ( IRB No. 2020-01-061)
Point 2: statistical analysis section: if p values of < 0.05 were considered significant, highlight them in some way in the tables (e.g. in bold).
Response 2: Thanks for your comment. As your comment, we highlighted the significant factor in table 4 using bold.

Reviewer 2 Report
It would be useful to provide a description/definition of stable and unstable SPLED. The reviewer recommends a minimum of 3 radiologists. The manuscript should indicate what is new in this work when compared with the current knowledge on the topic, is there any disagreement with what is correctly known? It would be useful to know, when combined with the findings in the paper, as the characteristics of the soft tissue injuries are interesting to be known, how the choice of treatment/rehabilitation can be affected? All figures should have a scale bar. Fig. 4. Captions should indicate what the arrow shows. It might be good to include the patient age and gender in the captions of the figures with MRI. Some of the references are old. Any key papers on the topic in the last 5 years should be cited if appropriate. ? All figures should have a scale bar.
Author Response
Please see the attachment below.

Round 2
Reviewer 2 Report
An optional minor suggestion, to enrich the discussion part of the paper, the authors may consider adapting and including some of their replies to the reviewers' previous comments in the manuscript. For instance the two parts about:
-the most important/new finding in the study
-the reflection of the results of the study on how the choice of treatment/rehabilitation can be affected
Author Response
An optional minor suggestion, to enrich the discussion part of the paper, the authors may consider adapting and including some of their replies to the reviewers' previous comments in the manuscript. For instance the two parts about:
-the most important/new finding in the study
-the reflection of the results of the study on how the choice of treatment/rehabilitation can be affected
Answer) Thanks for your comment. As your comment, we highlighted our result at the end of the discussion part.